# Knowledge Distillation Improves Graph Structure Augmentation for Graph Neural Networks

**Lirong Wu** [1,2], **Haitao Lin** [1,2], **Yufei Huang** [1,2], **and Stan Z. Li** [1†]

[1] AI Lab, School of Engineering, Westlake University
[2] College of Computer Science and Technology, Zhejiang University
`{wulirong,linhaitao,huangyufei,stan.zq.li}@westlake.edu.cn`

## Abstract

Graph (structure) augmentation aims to perturb the graph structure through heuristic or probabilistic rules, enabling the nodes to capture richer contextual information and thus improving generalization performance. While there have been a few graph structure augmentation methods proposed recently, none of them are aware of a potential *negative augmentation* problem, which may be caused by overly severe distribution shifts between the original and augmented graphs. In this paper, we take an important graph property, namely graph homophily, to analyze the distribution shifts between the two graphs and thus measure the severity of an augmentation algorithm suffering from negative augmentation. To tackle this problem, we propose a novel _Knowledge Distillation for Graph Augmentation_ (KDGA) framework, which helps to reduce the potential negative effects of distribution shifts, i.e., negative augmentation problem. Specifically, KDGA extracts the knowledge of any GNN teacher model trained on the augmented graphs and injects it into a partially parameter-shared student model that is tested on the original graph. As a simple but efficient framework, KDGA is applicable to a variety of existing graph augmentation methods and can significantly improve the performance of various GNN architectures. For three popular graph augmentation methods, namely GAUG, MH-Aug, and GraphAug, the experimental results show that the learned student models outperform their vanilla implementations by an average accuracy of 4.6% (GAUG), 4.2% (MH-Aug), and 4.6% (GraphAug) on eight graph datasets. Codes are available at: https://github.com/LirongWu/KDGA.

## 1 Introduction

In many real-world applications, including social networks, chemical molecules, and citation networks, data can be naturally modeled as graphs. Recently, the emerging Graph Neural Networks (GNNs) [4, 13, 48, 47, 22, 24, 28, 29] have demonstrated their powerful capability due to their superior performance in various graph-related tasks, including link prediction [59], node classification [21, 50], and graph classification [7]. Despite their great success, GNNs usually suffer from weak generalization due to its heavy reliance on the quantity of annotated labels and the quality of the graph structure. To boost generalization capabilities, a natural solution is to increase the amount of training data by creating plausible variations of existing data, which have been widely adopted in fields such as computer vision [33, 30, 9, 39, 25, 31, 26, 27] and natural language processing [46, 1, 36, 5, 34].

The data augmentation on graphs can be mainly divided into two branches: node feature augmentation and graph structure augmentation. While the former has been well studied by directly extending existing approaches for image and text data to graph data [56, 19, 15], comparatively little work has been done to study graph structure augmentation [35, 2, 62, 32]. Following the nomenclature of existing works [62, 32], we directly abbreviate *graph (structure) augmentation* to *graph augmentation*

36th Conference on Neural Information Processing Systems (NeurIPS 2022).

for the sake of brevity in this paper. The purpose of graph augmentation is to reasonably perturb the graph structure through heuristic or probabilistic rules, enabling the nodes to capture richer contextual information and thus improving generalization performance. For example, *DropEdge* [35] randomly removes a fraction of edges before each training epoch, in an approach reminiscent of dropout [40]. Besides, *AdaEdge* [2] iteratively adds (removes) edges between nodes predicted to have the same (different) labels with high confidence. In contrast to these heuristic methods, *GAUG* [62] proposes to optimize the graph augmentation and GNN parameters in an end-to-end manner. Similarly, *MH-Aug* [32] proposes a novel framework that draws a sequence of augmented graphs from an explicit target distribution, which enables flexible control of the strength and diversity of augmentation.

In this paper, we identify a potential *negative augmentation* problem for existing graph augmentation methods, i.e., the augmentation may cause overly severe *distribution shift* between the augmented graphs used for training and the original graph used for testing, which leads to suboptimal generalization. Moreover, we conduct extensive experiments to demonstrate the existence and hazard of distribution shifts and find that the direction of distribution shifts may be opposed on homophily and heterophily graphs. We propose a solution to the identified problem by adopting a _Knowledge Distillation for Graph Augmentation_ (KDGA) framework, which helps to reduce the potential negative effects of distribution shifts. Specifically, it extracts the knowledge of any GNN teacher model trained on the augmented graphs and injects it into a partially parameter-shared student model that is tested on the original graph. As a general framework, KDGA can significantly improve the vanilla implementations of various popular graph augmentation methods and GNN architectures.

Our contributions are summarized as follows: (1) We are the first to identify a potential negative augmentation problem for graph augmentation, and more importantly, we have described in detail what it represents, how it arises, what impact it has, and how to deal with it. (2) We proposes a novel _Knowledge Distillation for Graph Augmentation_ (KDGA) framework for the identified problem by directly distilling contextual information from augmented graphs. (3) We provide comprehensive experimental results showing that KDGA is applicable to a variety of graph augmentation methods and GNN models; it substantially outperforms the vanilla implementations across various datasets.

## 2 Background and Related Work

**Structure Augmentation for Graphs.** Data augmentation is an effective technique to improve generalization. Despite the great progress on node feature augmentation [56, 19, 15], comparatively little work study graph (structure) augmentation [35, 2, 62, 32, 6] due to the non-Euclidean property of structures. For graph data, the mainstream algorithms for structure augmentation are divided into two categories: heuristic and learning-based. As a typical heuristic algorithm, *DropEdge* [35] randomly remove edges according to the hand-crafted probability. In a similar way, *AdaEdge* [2] iteratively adds (removes) edges between nodes predicted to have the same (different) labels. Different from the above heuristic methods, *GAUG* [62] propose to optimize the graph augmentation and learnable GNN parameters in an end-to-end manner. Instead, *MH-Aug* [32] proposes a sampling-based augmentation, where a sequence of augmented graphs are directly drawn from an explicit target distribution.

**Graph Structure Learning and Graph Contrastive Learning.** Two closely related topics to graph augmentation are Graph Structure Learning [18, 23, 58, 20, 3] and Graph Contrastive Learning [19, 15, 57, 51, 63, 52], but *they are quite different in terms of learning objectives and evaluation protocols*. The learning goal of structure learning is to estimate a new structure with high quality [10, 8]. Instead, graph augmentation aims to reasonably perturb the graph structure during training to produce a set of augmented graphs, enabling nodes to receive richer contextual information; such augmentations allow the model to generalize better across those variations. As for the evaluation protocol, the augmented graphs are only used during training and are not available during testing. In contrast, for graph structure learning, the learned structure is used during both training and testing.

There are also some recent works [41, 65, 55] exploring how to perform data augmentation for graph contrastive learning, but they focus on automatically selecting the most appropriate transformations from a given pool to improve contrastive learning, rather than learning customized augmentation strategies for GNNs. More importantly, graph contrastive learning aims to learn transferable knowledge from abundant unlabeled data in an *unsupervised* setting and then generalize the learned knowledge to downstream tasks. Instead, graph augmentation usually works in a *semi-supervised*

setting, i.e., the label information is available during training. The graph structure learning and contrastive learning are not newly born topics, and we refer readers to the recent surveys [49, 64].

## 3 Preliminaries

**Notions.** Given a graph $\mathcal{G} = (\mathcal{V}, \mathcal{E})$, where $\mathcal{V}$ is the set of $N = |\mathcal{V}|$ nodes with features $\mathbf{X} = [\mathbf{x}_1, \mathbf{x}_2, \cdots, \mathbf{x}_N] \in \mathbb{R}^{N \times d}$ and $\mathcal{E}$ denotes the edge set. Each node $v_i \in \mathcal{V}$ is associated with a $d$-dimensional features vector $\mathbf{x}_i$, and each edge $e_{i,j} \in \mathcal{E}$ denotes a connection between node $v_i$ and $v_j$. The graph structure can also be denoted by an adjacency matrix $\mathbf{A} \in [0, 1]^{N \times N}$ with $\mathbf{A}_{i,j} = 1$ if $e_{i,j} \in \mathcal{E}$ and $\mathbf{A}_{i,j} = 0$ if $e_{i,j} \notin \mathcal{E}$. Consider a semi-supervised node classification task where only a subset of node $\mathcal{V}_L$ with corresponding labels $\mathcal{Y}_L$ are known, we denote the labeled set as $\mathcal{D}_L = (\mathcal{V}_L, \mathcal{Y}_L)$ and unlabeled set as $\mathcal{D}_U = (\mathcal{V}_U, \mathcal{Y}_U)$, where $\mathcal{V}_U = \mathcal{V} \backslash \mathcal{V}_L$. The node classification task aims to learn a mapping $\Phi : \mathcal{V} \rightarrow \mathcal{Y}$ on labeled data $\mathcal{D}_L$, so that it can be used to infer labels $\mathcal{Y}_U$.

**Background on Graph Homophily Ratio**. The homophily ratio is an important graph property that reflects the extent to which the graph structure adheres to the "label smoothness" criterion. The graph homophily ratio $r$ can be defined as the fraction of intra-class edges in the graph, as follows

$$r = \frac{|\{(i, j) : (i, j) \in \mathcal{E} \wedge y_i = y_j\}|}{|\mathcal{E}|} \tag{1}$$

where $y_i$ and $y_j$ are the ground-truth labels of node $v_i$ and $v_j$. In practice, the distribution space size of a discrete graph structure $\mathbf{A} \in [0, 1]^{N \times N}$ is $2^{N^2}$, making it tractable to directly estimate the distribution differences between two discrete graph structures. In this paper, we take the graph homophily as a desirable option to analyze the distribution shifts between the original and augmented graphs, thus measuring the severity of an algorithm suffering from the *negative augmentation* problem.

## 4 Methodology

In this section, we first make problem statements for graph augmentation in Sec. 4.1, highlight our motivations by analyzing the distribution shift between the original and augmented graphs in Sec. 4.2, then present a novel teacher-student $\underline{K}$*nowledge* $\underline{D}$*istillation for* $\underline{G}$*raph* $\underline{A}$*ugmentation* (KDGA) framework in Sec. 4.3, and finally provide one of its specific instantiations in Sec. 4.4.

### 4.1 Problem Statement

**Graph Representation Learning.** From the perspective of statistical learning, the key of node classification is to learn a mapping $p(Y \mid \mathbf{X}, \mathbf{A})$ based on node features $\mathbf{X}$ and graph structure $\mathbf{A}$. The learned mapping can be used to infer labels $\mathcal{Y}_U$ on the graph structure $\mathbf{A}$ as shown in Fig. 1(a).

**Graph Structure Learning.** The goal of graph structure learning is to estimate a more accurate structure $\widehat{\mathbf{A}}$ by another mapping $p(\widehat{\mathbf{A}} \mid \mathbf{X}, \mathbf{A})$ and then feed it into the mapping $p(Y \mid \mathbf{X}, \widehat{\mathbf{A}})$ along with node features $\mathbf{X}$. Finally, the learned mapping $p(Y \mid \mathbf{X}, \widehat{\mathbf{A}})$ can be used to infer labels $\mathcal{Y}_U$ on the estimated (high-quality) structure $\widehat{\mathbf{A}}$ instead of the original strucute $\mathbf{A}$ as shown in Fig. 1(b).

**Graph Augmentation.** Instead of directly working with the original graph, we would like to leverage graph augmentation to reasonably perturb the graph structure and learn more generalizable representations. In other words, we are interested in the following variant, as follows

$$p(Y \mid \mathbf{X}, \mathbf{A}) = \sum_{\widehat{\mathbf{A}} \in [0,1]^{N \times N}} p(Y \mid \mathbf{X}, \widehat{\mathbf{A}}) p(\widehat{\mathbf{A}} \mid \mathbf{X}, \mathbf{A}) \tag{2}$$

where $\widehat{\mathbf{A}} \in [0, 1]^{N \times N}$ is the augmented graph (structure). In practice, the distribution space size of $\widehat{\mathbf{A}}$ is $2^{N^2}$, and it is intractable to enumerate all possible $\widehat{\mathbf{A}}$ as well as estimate the exact values of the mappings $p(Y \mid \mathbf{X}, \widehat{\mathbf{A}})$ and $p(\widehat{\mathbf{A}} \mid \mathbf{X}, \mathbf{A})$. Therefore, we approximate them by tractable functions as

$$p(Y \mid \mathbf{X}, \mathbf{A}) = \sum_{\widehat{\mathbf{A}} \in [0,1]^{N \times N}} q_\theta(Y \mid \mathbf{X}, \widehat{\mathbf{A}}) q_\phi(\widehat{\mathbf{A}} \mid \mathbf{X}, \mathbf{A}) \tag{3}$$

where $q_\theta(\cdot)$ and $q_\phi(\cdot)$ are approximation functions for $p(Y \mid \mathbf{X}, \widehat{\mathbf{A}})$ and $p(\widehat{\mathbf{A}} \mid \mathbf{X}, \mathbf{A})$ parameterized by $\theta$ and $\phi$, respectively. In practice, the function $q_\theta(Y \mid \mathbf{X}, \widehat{\mathbf{A}})$ can be generally implemented by

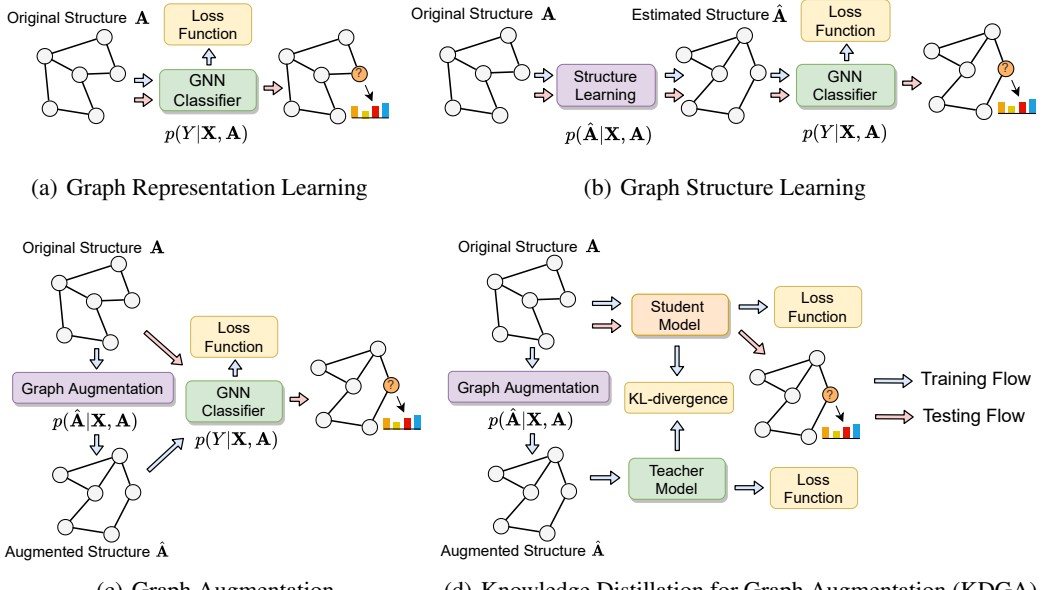

(a) Graph Representation Learning $\qquad$ (b) Graph Structure Learning

(c) Graph Augmentation $\qquad$ (d) Knowledge Distillation for Graph Augmentation (KDGA)

Figure 1: Illustrations of graph representation learning, graph structure learning, graph augmentation, and the proposed KDGA framework. For the sake of chart brevity, we omitted the node features $\mathbf{X}$.

GNNs, and the function $q_\phi(\widehat{\mathbf{A}} \mid \mathbf{X}, \mathbf{A})$ can be implemented by graph augmentation methods to model the distributions of augmented graph structures. Once the model training is finished, the mapping $q_\theta(Y \mid \mathbf{X}, \mathbf{A})$ can be used to infer labels $\mathcal{Y}_U$ on the original structure $\mathbf{A}$ as shown in Fig. 1(c).

In summary, unlike graph representation and graph structure learning that leverage the same structure ($\mathbf{A}$ or $\widehat{\mathbf{A}}$) for both training and testing, the graph structures for training and testing in graph augmentation are completely different, which may lead to a potential negative augmentation problem.

### 4.2 Motivation: Potential Negative Augmentation Problem

One may create a model by specifying specific implementations for functions $q_\theta(Y \mid \mathbf{X}, \widehat{\mathbf{A}})$ and $q_\phi(\widehat{\mathbf{A}} \mid \mathbf{X}, \mathbf{A})$ and then optimize it by maximizing the posterior $p(Y \mid \mathbf{X}, \mathbf{A})$ defined in Eq. (3). As we will explain here, however, this model may suffer from a potential negative augmentation problem caused by overly severe distribution shifts between the original and augmented graphs.

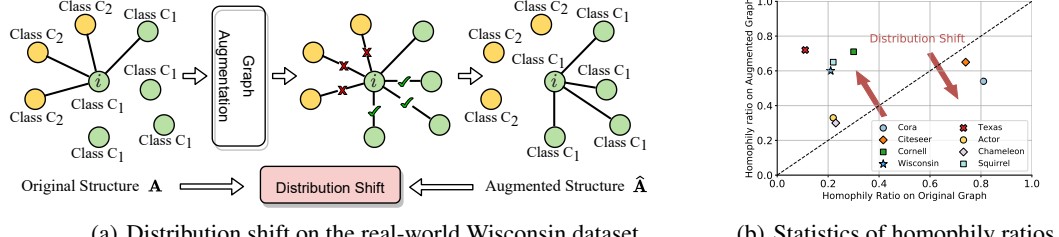

(a) Distribution shift on the real-world Wisconsin dataset $\qquad$ (b) Statistics of homophily ratios

Figure 2: Illustrations of how the distribution shift is arising and how it behaves on different datasets.

The distribution shift itself is not necessarily harmful; it is actually a neutral phenomenon. A proper distribution shift helps the model "see" more different graphs, enabling the nodes to receive more contextual information, thus improving generalization; however, an overly severe distribution shift can lead to a potential negative augmentation problem. To illustrate it, we consider a node $v_i$ (id 129) of class $C_1$ from the real-world Wisconsin dataset in Fig. 2(a), it is initially connected to a node with the same class $C_1$ and three nodes from another class $C_2$ in the original structure $\mathbf{A}$. During the training process, the original structure $\mathbf{A}$ and node features $\mathbf{X}$ are fed together into $q_\phi(\widehat{\mathbf{A}} \mid \mathbf{X}, \mathbf{A})$ to generate an augmented structure $\widehat{\mathbf{A}}$. Under the downstream supervision, it disconnects from three nodes from class $C_2$ and reconnects with three nodes from the same class $C_1$, resulting in an

augmented structure $\widehat{\mathbf{A}}$ with a much higher homophily ratio, that is, an overly severe distribution shift between the original and augmented graphs from the perspective of graph homophily property. As a result, a model trained on the augmented structure $\widehat{\mathbf{A}}$ can successfully predict node $i$ as class $C_1$, but make a wrong prediction $C_2$ for node $i$ when tested on the original structure $\mathbf{A}$, which is termed as *"negative augmentation"*. Furthermore, we plot the homophily ratios of the original structure $\mathbf{A}$ and augmented structure $\widehat{\mathbf{A}}$ on eight datasets in Fig. 2(b), from which we can observe significant distribution shifts between the two graphs. Moreover, while the above analysis is developed on a heterophily (Wisconsin) graph, we find that the identified distributional shift also exists in homophily graphs, only in a different direction. Please see Sec. 5.3 for detailed experimental settings and results.

## 4.3 Knowledge Distillation for Graph Augmentation (KDGA)

The distribution shift is essentially a trade-off between better generalizability and higher risks of negative augmentation. However, the *optimal* distribution shift may vary from dataset to dataset, or even from node to node, making it challenging to directly control the levels of distribution shifts. In this paper, we have not attempted to control or prevent distribution shifts. Instead, we allow for the existence of any level of distribution shifts, but we reduce their negative impact, i.e., the potential negative augmentation problem, by the proposed KDGA framework, which gradually distills the contextual information from the augmented graphs into a student model tested on the original graph. The idea of KDGA is straightforward, yet as we will see, extremely effective. In our case, we first generate soft distributions $\mathbf{z}_i^T$ and $\mathbf{z}_i^S$ for node $v_i$ with the teacher and student models, respectively. The knowledge distillation is first introduced in [14], where knowledge was transferred from a cumbersome teacher to a simpler student by optimizing the following objective function, as follows

$$\mathcal{L}_{\mathrm{KD}} = \frac{1}{|\mathcal{V}|} \sum_{i \in \mathcal{V}} \mathcal{D}_{KL} \left( \mathrm{softmax}\left(\mathbf{z}_i^T\right), \mathrm{softmax}\left(\mathbf{z}_i^S\right) \right) \tag{4}$$

In this paper, not to get a simpler student model, we adopt the knowledge distillation framework to address the identified negative augmentation problem caused by overly severe distribution shifts between the two graphs. In short, we extract the knowledge of any teacher model trained on the augmented graphs and inject it into a student model tested on the original graph as in Fig. 1(d).

**Teacher Model.** The teacher model can be implemented by any GNN, which takes node features $\mathbf{X}$ and augmented structure $\widehat{\mathbf{A}}$ as input and learn latent node representations via neighborhood feature aggregation. Considering a $L$-layer GNN $f_\theta(\mathbf{X}, \widehat{\mathbf{A}})$, the formulation of the $l$-th layer is as follows

$$\mathbf{h}_{i,T}^{(l+1)} = \mathrm{UPDATE}^{(l)}\left(\mathbf{h}_{i,T}^{(l)}, \mathrm{AGGREGATE}^{(l)}\left(\left\{\mathbf{h}_{j,T}^{(l)} : v_j \in \mathcal{N}_i^{\widehat{\mathbf{A}}}\right\}\right)\right) \tag{5}$$

where $0 \le l \le L-1$, $\mathbf{h}_{i,T}^{(0)} = \mathbf{x}_i$ is the input feature, and $\mathcal{N}_i^{\widehat{\mathbf{A}}}$ is the neighborhood of node $v_i$ in the augmented structure $\widehat{\mathbf{A}}$. After $L$ message-passing layers, the final node embedding $\mathbf{h}_{i,T}^{(L)}$ is passed to a linear prediction head $g^T(\cdot)$ to obtain logits $\mathbf{z}_i^T = g^T(\mathbf{h}_{i,T}^{(L)})$, and the model is trained by a cross-entropy loss $\mathcal{H}(\cdot)$ with ground-truth labels $\mathcal{Y}_L$, given by $\mathcal{L}_{SUP}^T = \sum_{i \in \mathcal{V}_L} \mathcal{H}\left(y_i; \mathrm{softmax}\left(\mathbf{z}_i^T\right)\right)$.

**Student Model.** The student model $f_\theta(\mathbf{X}, \mathbf{A})$ shares the parameters $\theta$ with the teacher model, but differs in that the it takes the original structure $\mathbf{A}$ as input, as shown in Fig. 3(c). Besides, an additional linear prediction head $g^S(\cdot)$ is used to map the node embedding $\mathbf{h}_{i,S}^{(L)}$ to logits $\mathbf{z}_i^S = g^S(\mathbf{h}_{i,S}^{(L)})$. As already explained earlier, the augmented graphs enable the teacher model to receive richer contextual information, which helps to improve model generalization. To allow the student model tested on the original structure to also benefit from it, we consider the contextual neighborhood information from both original and augmented structures and distill them into the student model, defined as

$$\mathcal{L}_{\mathrm{GKD}} = \frac{\tau_1^2}{|\mathcal{V}|} \sum_{i \in \mathcal{V}} \sum_{j \in (\mathcal{N}_i^{\mathbf{A}} \cap \mathcal{N}_i^{\widehat{\mathbf{A}}}) \cup i} \mathcal{D}_{KL}\left(\mathrm{softmax}\left(\mathbf{z}_j^T / \tau_1\right), \mathrm{softmax}\left(\mathbf{z}_i^S / \tau_1\right)\right) \tag{6}$$

where $\tau_1$ is the distillation temperature, and $\tau_1^2$ is used to keep the gradient stability of this loss [14].

**Discussions.** While a large number of methods on graph knowledge distillation [61, 53, 60] have been proposed, most of them adopt the standard teacher-student knowledge distillation framework

as shown in Fig. 3(a), where the inputs to both teacher and student models are the same (structure). Despite many progresses, their contributions have mostly focused on the special design of the teacher or student models. For example, CPF [54] proposes to distill knowledge from a teacher GNN to a student MLP, but it specifically incorporates label propagation [16] into the student model to improve performance. In contrast, GDK [11] utilizes label propagation in the teacher model to fully exploit both feature and topological information. In our proposed KDGA framework, the graph structures fed to the teacher and student models are completely different. Moreover, unlike the scheme in Fig. 3(b) where two parameter-independent teacher and student models are used, we adopt the architecture shown in Fig. 3(c) where the GNN parameters are shared but with two independent prediction heads to increase discriminability. The behind motivation is that a parameter-independent student model has the risk of quickly fitting with the original structure under the optimization of downstream supervision, while failing to take full advantage of rich contextual information from the augmented graphs. A detailed comparison of parameter-independent and parameter-shared schemes is reported in Table. 2.

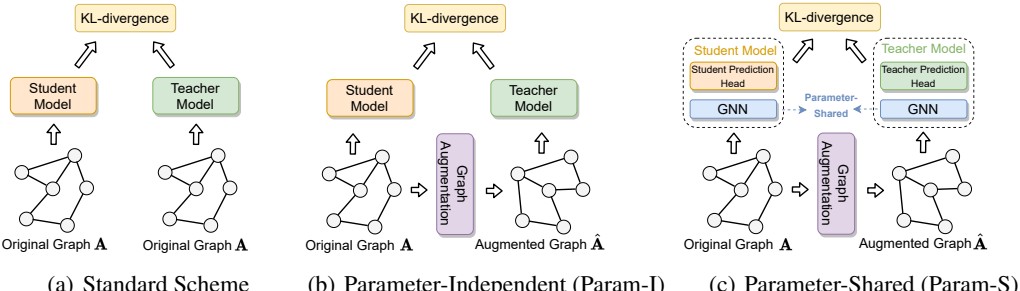

(a) Standard Scheme      (b) Parameter-Independent (Param-I)      (c) Parameter-Shared (Param-S)

Figure 3: (a) standard teacher-student distillation; (b) distillation with parameter-independent teacher and student models; (c) distillation with parameter-shared teacher and student models.

## 4.4 Instantiating KDGA with GraphAug

In practice, any existing graph augmentation method can be used to instantiate the proposed KDGA framework and achieve consistent improvements over the vanilla implementations, as shown in Table. 1. In this subsection, we adopt a probabilistic generative-based graph augmentation method to model the function $q_\phi(\widehat{\mathbf{A}} \mid \mathbf{X}, \mathbf{A})$, termed GraphAug, and use it to instantiate our KDGA framework. Specifically, we introduce a set of discrete variables $\Lambda = \{\lambda_{i,j}\}_{i,j=1}^N$ to model the distribution of the augmented graph, where $\lambda_{i,j} \in \{0,1\}$ denotes the augmentation probability between node $v_i$ and $v_j$. Moreover, we avoid estimating the probability $p(\lambda_{i,j} \mid \mu_{i,j})$ using independent local parameter $\mu_{i,j}$ and instead fits a shared neural network to estimate it. Specifically, we first transform the input to a low-dimensional hidden space, done by multiplying the node features with a parameter matrix $\mathbf{W} \in \mathbb{R}^{F \times d}$, that is, $\mathbf{e}_i = \mathbf{W}x_i$. Then, we directly parameterize the probability $\lambda_{i,j}$ as

$$p(\lambda_{i,j} \mid \mathbf{X}, \mathbf{A}) = \sigma\left(\mathbf{e}_i \mathbf{e}_j^T\right) \tag{7}$$

where $\sigma(\cdot)$ is an element-wise sigmoid function. Next, to sample discrete augmented graphs from the learned augmentation distribution and make the sampling process differentiable, we adopt Gumbel-Softmax sampling [17]. Specifically, the sampling process can be formulated as

$$\widehat{\mathbf{A}}_{i,j} = \left\lfloor \frac{1}{1 + \exp^{-\left(\log \mathbf{M}_{i,j} + G\right)/\tau_2}} + \frac{1}{2} \right\rfloor, \text{where} \quad \mathbf{M}_{i,j} = \alpha p(\lambda_{i,j} \mid \mathbf{X}, \mathbf{A}) + (1 - \alpha)\mathbf{A}_{i,j} \tag{8}$$

where $\alpha \in [0,1]$ is the fusion factor to control the intensity of the graph augmentation, $\tau_2$ is the augmentation temperature, and $G \sim \text{Gumbel}(0,1)$ is a gumbel random variate.

To warm-up the proposed GraphAug module, we first pre-train it with loss $\mathcal{L}_{Aug} = \frac{1}{N^2}\mathcal{H}(\mathbf{A}_{i,j}, \widehat{\mathbf{A}}_{i,j})$, where $\mathcal{H}(\cdot)$ denotes the cross-entropy loss. Besides, we use classification loss $\mathcal{L}_{Cla} = \frac{1}{|\mathcal{V}_L|}\sum_{i \in \mathcal{V}_L}\left(\mathcal{H}\left(y_i, \text{softmax}(\mathbf{z}_i^T)\right) + \mathcal{H}\left(y_i, \text{softmax}(\mathbf{z}_i^S)\right)\right)$ to pre-train the teacher and student models until it converges. Finally, the total loss to train the whole framework is defined as follows

$$\mathcal{L}_{total} = \mathcal{L}_{Cla} + \lambda\mathcal{L}_{Aug} + \kappa\mathcal{L}_{GKD} \tag{9}$$

where $\lambda$ and $\kappa$ are the weights to balance the influence of the two losses $\mathcal{L}_{Aug}$ and $\mathcal{L}_{GKD}$.

Table 1: Accuracy $\pm$ std (%) on eight datasets (as well as their homophily ratios), with three GNN architectures and five graph augmentation methods considered. The best metrics are marked by **bold**.

| BaseGNN | Method | Cora | Citeseer | Cornell | Chameleon | Squirrel | Actor | Wisconsin | Texas |
|---|---|---|---|---|---|---|---|---|---|
| | | 0.81 | 0.74 | 0.30 | 0.23 | 0.22 | 0.22 | 0.21 | 0.11 |
| GCN | Vanilla | 81.5$_{\pm0.8}$ | 71.6$_{\pm0.3}$ | 57.0$_{\pm4.7}$ | 59.8$_{\pm2.6}$ | 36.9$_{\pm1.3}$ | 30.3$_{\pm0.8}$ | 59.8$_{\pm7.0}$ | 59.5$_{\pm5.3}$ |
| | DropEdge [35] | 82.2$_{\pm0.7}$ | 71.9$_{\pm0.3}$ | 59.3$_{\pm3.9}$ | 61.2$_{\pm1.8}$ | 38.1$_{\pm1.5}$ | 30.9$_{\pm1.0}$ | 61.8$_{\pm5.4}$ | 62.3$_{\pm4.6}$ |
| | AdaEdge [2] | 82.3$_{\pm0.8}$ | 69.7$_{\pm0.9}$ | 57.8$_{\pm4.3}$ | 59.5$_{\pm2.3}$ | 37.6$_{\pm1.4}$ | 31.4$_{\pm1.2}$ | 60.4$_{\pm4.7}$ | 58.8$_{\pm4.0}$ |
| | SSL [63] | 83.8$_{\pm0.7}$ | 72.9$_{\pm0.6}$ | 58.8$_{\pm3.2}$ | 60.4$_{\pm2.1}$ | 39.5$_{\pm1.9}$ | 30.5$_{\pm1.2}$ | 62.8$_{\pm4.5}$ | 63.3$_{\pm4.6}$ |
| | GraphMix [44] | 83.9$_{\pm0.6}$ | **74.7$_{\pm0.6}$** | 60.5$_{\pm3.7}$ | 61.2$_{\pm2.3}$ | 41.1$_{\pm1.5}$ | 31.4$_{\pm0.9}$ | 62.4$_{\pm5.0}$ | 62.3$_{\pm4.6}$ |
| | GAUG [62] | 83.6$_{\pm0.5}$ | 73.3$_{\pm1.1}$ | 55.8$_{\pm4.0}$ | 59.3$_{\pm1.4}$ | 36.3$_{\pm0.8}$ | 29.7$_{\pm0.9}$ | 57.5$_{\pm5.1}$ | 58.0$_{\pm4.2}$ |
| | GAUG (w/ KDGA) | **85.4$_{\pm0.7}$** | 73.6$_{\pm0.6}$ | 63.2$_{\pm3.6}$ | 63.0$_{\pm1.2}$ | 46.2$_{\pm0.9}$ | 33.3$_{\pm0.8}$ | 65.0$_{\pm2.5}$ | 67.4$_{\pm3.8}$ |
| | $\Delta_{Acc}$ | 1.8 | 0.3 | 7.4 | 3.7 | 9.9 | 3.6 | 7.5 | 9.4 |
| | MH-Aug [32] | 83.6$_{\pm0.3}$ | 73.0$_{\pm0.5}$ | 58.4$_{\pm3.5}$ | 59.2$_{\pm2.0}$ | 35.9$_{\pm1.0}$ | 31.2$_{\pm0.7}$ | 58.1$_{\pm5.3}$ | 58.9$_{\pm3.9}$ |
| | MH-Aug (w/ KDGA) | 85.0$_{\pm0.5}$ | 73.8$_{\pm0.8}$ | 63.5$_{\pm2.7}$ | **63.3$_{\pm1.7}$** | 45.4$_{\pm1.1}$ | **34.8$_{\pm1.0}$** | 65.7$_{\pm2.7}$ | 67.2$_{\pm2.6}$ |
| | $\Delta_{Acc}$ | 1.4 | 0.8 | 5.1 | 4.1 | 9.5 | 3.6 | 7.6 | 8.3 |
| | GraphAug | 83.2$_{\pm0.9}$ | 73.2$_{\pm0.8}$ | 56.6$_{\pm2.4}$ | 58.8$_{\pm1.8}$ | 37.2$_{\pm1.2}$ | 28.8$_{\pm0.9}$ | 59.3$_{\pm2.6}$ | 59.4$_{\pm3.3}$ |
| | GraphAug (w/ KDGA) | 85.2$_{\pm0.7}$ | 73.9$_{\pm0.7}$ | **63.8$_{\pm3.2}$** | 62.7$_{\pm1.5}$ | **46.9$_{\pm0.6}$** | 32.5$_{\pm0.6}$ | **66.3$_{\pm1.9}$** | **68.0$_{\pm2.3}$** |
| | $\Delta_{Acc}$ | 2.0 | 0.7 | 7.2 | 3.9 | 9.7 | 3.7 | 6.9 | 8.6 |
| SAGE | Vanilla | 79.8$_{\pm0.7}$ | 71.1$_{\pm0.6}$ | 76.0$_{\pm5.0}$ | 58.7$_{\pm1.7}$ | 41.6$_{\pm0.7}$ | 34.2$_{\pm1.0}$ | 81.2$_{\pm5.6}$ | 82.4$_{\pm6.1}$ |
| | DropEdge [35] | 80.4$_{\pm0.8}$ | 71.5$_{\pm0.6}$ | 77.4$_{\pm3.6}$ | 60.2$_{\pm2.0}$ | 42.5$_{\pm1.3}$ | 36.4$_{\pm1.3}$ | 82.7$_{\pm4.4}$ | 83.0$_{\pm4.8}$ |
| | AdaEdge [2] | 80.2$_{\pm1.2}$ | 69.4$_{\pm0.8}$ | 76.5$_{\pm4.6}$ | 59.5$_{\pm1.6}$ | 40.3$_{\pm1.6}$ | 34.9$_{\pm0.8}$ | 82.0$_{\pm5.3}$ | 81.6$_{\pm5.3}$ |
| | SSL [63] | 82.5$_{\pm0.8}$ | 71.2$_{\pm0.5}$ | 76.8$_{\pm3.4}$ | 59.1$_{\pm1.8}$ | 42.0$_{\pm1.5}$ | 35.2$_{\pm1.2}$ | 82.4$_{\pm3.6}$ | 82.6$_{\pm4.4}$ |
| | GraphMix [44] | 82.3$_{\pm0.6}$ | 69.6$_{\pm0.4}$ | 78.0$_{\pm4.2}$ | 59.9$_{\pm2.0}$ | 42.6$_{\pm1.6}$ | 35.8$_{\pm1.0}$ | 83.1$_{\pm4.1}$ | 83.5$_{\pm3.9}$ |
| | GAUG [62] | 82.0$_{\pm0.5}$ | 72.7$_{\pm0.7}$ | 74.8$_{\pm4.2}$ | 58.2$_{\pm1.3}$ | 40.5$_{\pm0.9}$ | 34.4$_{\pm1.1}$ | 80.7$_{\pm4.6}$ | 82.0$_{\pm4.5}$ |
| | GAUG (w/ KDGA) | 84.5$_{\pm0.8}$ | 73.4$_{\pm0.7}$ | 80.6$_{\pm3.5}$ | 61.8$_{\pm1.4}$ | **46.4$_{\pm1.1}$** | 36.4$_{\pm0.7}$ | **85.5$_{\pm3.2}$** | 84.5$_{\pm3.6}$ |
| | $\Delta_{Acc}$ | 2.5 | 0.7 | 5.8 | 3.6 | 5.9 | 2.0 | 4.8 | 2.5 |
| | MH-Aug [32] | 82.6$_{\pm0.7}$ | 72.1$_{\pm1.0}$ | 75.3$_{\pm3.9}$ | 59.4$_{\pm1.5}$ | 41.0$_{\pm0.8}$ | 33.8$_{\pm0.8}$ | 80.5$_{\pm5.0}$ | 81.2$_{\pm5.2}$ |
| | MH-Aug (w/ KDGA) | 84.3$_{\pm0.7}$ | **73.7$_{\pm0.8}$** | 80.3$_{\pm3.2}$ | **62.1$_{\pm1.3}$** | 45.9$_{\pm1.4}$ | 35.9$_{\pm0.7}$ | 84.9$_{\pm4.0}$ | 83.8$_{\pm4.4}$ |
| | $\Delta_{Acc}$ | 1.7 | 1.6 | 5.0 | 2.7 | 4.9 | 2.1 | 4.4 | 2.6 |
| | GraphAug | 82.4$_{\pm1.0}$ | 72.4$_{\pm0.9}$ | 75.8$_{\pm3.0}$ | 58.8$_{\pm1.4}$ | 40.2$_{\pm1.3}$ | 33.2$_{\pm0.7}$ | 79.9$_{\pm4.2}$ | 81.9$_{\pm4.6}$ |
| | GraphAug (w/ KDGA) | **84.8$_{\pm0.8}$** | 73.5$_{\pm0.5}$ | **81.4$_{\pm2.8}$** | 61.0$_{\pm1.8}$ | 45.6$_{\pm0.9}$ | **36.9$_{\pm1.4}$** | 84.5$_{\pm3.3}$ | **84.8$_{\pm3.8}$** |
| | $\Delta_{Acc}$ | 2.4 | 1.1 | 5.6 | 2.2 | 5.4 | 3.7 | 4.6 | 2.9 |
| GAT | Vanilla | 82.2$_{\pm0.5}$ | 71.4$_{\pm0.9}$ | 58.9$_{\pm3.3}$ | 54.7$_{\pm2.0}$ | 30.6$_{\pm2.1}$ | 26.3$_{\pm1.7}$ | 55.3$_{\pm8.7}$ | 58.4$_{\pm4.5}$ |
| | DropEdge [35] | 83.0$_{\pm0.4}$ | 72.2$_{\pm0.9}$ | 60.2$_{\pm3.8}$ | 55.6$_{\pm2.5}$ | 34.1$_{\pm1.7}$ | 28.2$_{\pm1.5}$ | 57.8$_{\pm5.5}$ | 60.5$_{\pm3.8}$ |
| | DropEdge [35] | 77.9$_{\pm2.0}$ | 69.1$_{\pm0.8}$ | 57.7$_{\pm4.5}$ | 54.0$_{\pm2.2}$ | 32.8$_{\pm2.0}$ | 27.5$_{\pm1.4}$ | 56.4$_{\pm6.1}$ | 57.8$_{\pm4.2}$ |
| | SSL [63] | 83.7$_{\pm0.6}$ | 72.7$_{\pm0.7}$ | 60.6$_{\pm3.2}$ | 55.8$_{\pm2.2}$ | 35.0$_{\pm1.3}$ | 27.6$_{\pm1.3}$ | 57.2$_{\pm5.1}$ | 60.5$_{\pm3.3}$ |
| | GraphMix [44] | 83.3$_{\pm0.2}$ | 73.1$_{\pm0.2}$ | 61.0$_{\pm4.1}$ | 56.4$_{\pm1.7}$ | 35.6$_{\pm1.0}$ | 28.7$_{\pm0.9}$ | 58.5$_{\pm4.5}$ | 61.1$_{\pm2.8}$ |
| | GAUG [62] | 82.2$_{\pm0.8}$ | 71.6$_{\pm1.1}$ | 57.6$_{\pm3.8}$ | 53.4$_{\pm1.4}$ | 30.1$_{\pm1.5}$ | 25.8$_{\pm1.0}$ | 54.8$_{\pm5.7}$ | 56.9$_{\pm3.6}$ |
| | GAUG (w/ KDGA) | 84.2$_{\pm1.1}$ | 73.0$_{\pm0.7}$ | 62.2$_{\pm3.4}$ | 58.2$_{\pm1.1}$ | **39.1$_{\pm1.3}$** | **31.3$_{\pm1.2}$** | 60.9$_{\pm5.3}$ | 63.1$_{\pm3.2}$ |
| | $\Delta_{Acc}$ | 2.0 | 1.4 | 4.6 | 4.8 | 9.0 | 5.5 | 6.1 | 6.2 |
| | MH-Aug [32] | 83.5$_{\pm0.7}$ | 72.8$_{\pm1.0}$ | 58.0$_{\pm4.0}$ | 55.3$_{\pm1.8}$ | 29.5$_{\pm1.1}$ | 25.7$_{\pm1.2}$ | 55.8$_{\pm4.0}$ | 57.8$_{\pm4.0}$ |
| | MH-Aug (w/ KDGA) | 84.5$_{\pm0.9}$ | **73.4$_{\pm0.8}$** | 62.7$_{\pm2.8}$ | **59.5$_{\pm1.6}$** | 37.3$_{\pm0.8}$ | 30.8$_{\pm0.9}$ | 61.4$_{\pm5.0}$ | **64.4$_{\pm2.8}$** |
| | $\Delta_{Acc}$ | 1.0 | 0.6 | 4.7 | 4.2 | 7.8 | 5.1 | 5.6 | 6.6 |
| | GraphAug | 83.2$_{\pm0.8}$ | 72.5$_{\pm0.7}$ | 58.6$_{\pm3.4}$ | 54.0$_{\pm1.7}$ | 29.8$_{\pm1.6}$ | 24.8$_{\pm1.3}$ | 54.4$_{\pm3.6}$ | 57.1$_{\pm4.4}$ |
| | GraphAug (w/ KDGA) | **84.7$_{\pm0.7}$** | 73.2$_{\pm0.8}$ | **63.1$_{\pm2.5}$** | 58.8$_{\pm1.3}$ | 38.9$_{\pm1.4}$ | 30.0$_{\pm1.0}$ | **61.8$_{\pm4.7}$** | 62.7$_{\pm2.0}$ |
| | $\Delta_{Acc}$ | 1.5 | 0.7 | 4.5 | 4.8 | 9.1 | 5.2 | 7.4 | 5.6 |

# 5 Experiments

**Datasets.** The effectiveness of the proposed KDGA framework is evaluated on *eight* datasets. We use two commonly used homophily graph datasets, namely *Cora* [38] and *Citeseer* [12] as well as six heterophily graph datasets: *Cornell, Texas, Wisconsin*, *Aactor* [42], *Chameleon* and *Squirrel* [37]. A statistical overview of these datasets is available in **Appendix A**. We defer the implementation details and the best hyperparameter settings for each dataset to **Appendix B** and supplementary material.

**Baselines.** As a general framework, KDGA can be combined with any GNN architecture and existing graph augmentation methods. In this paper, we consider three GNN architectures, GCN [21], GraphSAGE [13], and GAT [43]. Besides, to demonstrate the applicability of KDGA to various graph augmentation methods in addition to the proposed GraphAug, we also consider two state-of-the-art learning-based baselines, GAUG [62] and MH-Aug [32]. In particular, two heuristics methods, DropEdge and AdaEdge, are also included in the comparison as baselines. Moreover, we also compare KDGA with two semi-supervised methods: (1) GraphMix [44], a regularization method that performs linear interpolation between two data on graphs, and (2) SSL [63], that proposes two self-supervised tasks to fully exploit available information embedded in the graph structure. Each set of experiments is run five times with different random seeds, and the average performance is reported.

## 5.1 Comparative Results

To evaluate the powerful capabilities of the proposed KDGA framework, we instantiate it with three learning-based graph augmentation methods, GAUG, MH-Aug, and GraphAug. The experiments are conducted on eight datasets with three different GNN architectures. From the experimental results shown in Table. 1, we can make the following observations: (1) Two heuristic graph augmentation methods, DropEdge and AdaEdge, can improve the performance of the vanilla GNNs overall. However, such improvements are usually very limited and do not work for all datasets and GNN architectures. For example, on the Citeseer dataset, the performance of AdaEdge drops over the vanilla GNNs by 1.9% (GCN), 1.7% (GraphSAGE), and 2.3% (GAT), respectively. (2) There are huge gaps in the effectiveness of three learning-based augmentation methods on homophily and heterophily graphs. While these methods can significantly improve performance on homophily graphs, their performance gains on heterophily graphs are greatly reduced and even detrimental. For example, with GCN as the GNN architecture, the performance of GAUG improves by 2.1% on Cora, but drops by 1.5% and 1.2% on Texas and Cornell. Such negative augmentation is mainly caused by the overly severe distribution shift between the original and augmented graphs as analyzed in Sec. 4.2. (3) The proposed KDGA framework can consistently improve the performance of vanilla graph augmentation methods across three GNN architectures on all eight datasets, especially for those heterophily graphs. For example, with GCN as the GNN architecture, the performance of GraphAug can be improved by 9.7% and 8.6% on the Squirrel and Texas datasets. (4) Two semi-supervised approaches, SSL and GraphMix, can achieve comparable or even better performance than learning-based graph augmentation, especially on heterophily graphs. However, by combining with KDGA, GAUG, MH-Aug, and GraphAug outperform both SSL and GraphMix by a large margin overall.

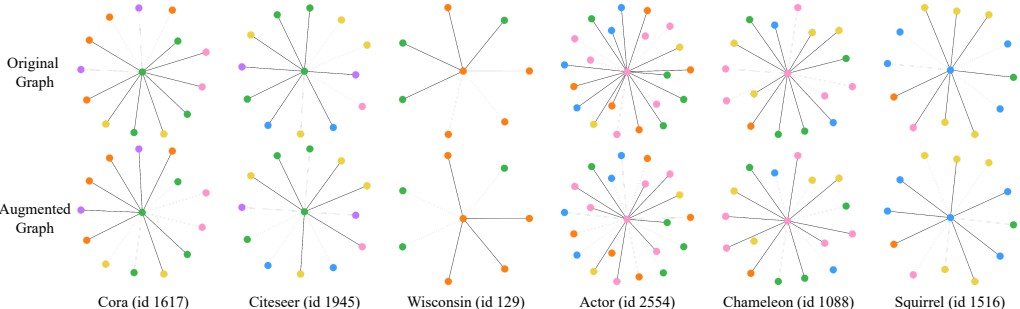

Figure 4: Case studies for each dataset, where we pick a node with the most drastic neighborhood variations and visualize its neighborhood on the original graph structure (top) and augmented graph structure (bottom), where each node is colored according to its ground-truth label.

## 5.2 Analysis on the Distribution Shift and Negative Augmentation

Next, we qualitatively and quantitatively analyze the distribution shift between the original and augmented graphs and explain how it can cause a potential negative augmentation. Without loss of generality, we consider GCN as the GNN architecture and GraphAug as the augmentation method.

**Visualizations of Neighborhood Variations.** First, we pick a node with the most drastic neighborhood variations and visualize its neighborhood of the original and augmented graphs in Fig. 4, where each node is colored according to its ground-truth label. The visualizations show that there would be a huge gap between the neighborhoods of the original and augmented graphs, which causes a model that is well trained on the augmented graph to predict poorly on the original graph during testing. Taking the Wisconsin dataset as an example, the selected node is connected to four nodes from the same class in the augmented graph, so it can be well trained to make correct predictions. However, its neighborhood context is completely changed in the original graph, where the node is connected to three nodes from different classes, so it will be predicted with high confidence to an incorrect class.

**Training Curves of Homophily Ratios.** We plot in Fig. 5 the training curves (w/o GKD Loss) of the homophily ratios of the original and augmented graphs during training. It can be seen that their gaps are enlarged as training proceeds, which indicates that the distribution of the augmented graphs is gradually shifting from the original graph. This shift may even reach 0.5 for some datasets (e.g., Texas), in which case the graph homophily property is completely reversed. More importantly, we find that the direction of distribution shifts may be completely opposite for homophily and heterophily graphs, which makes it more challenging to solve the negative augmentation problem.

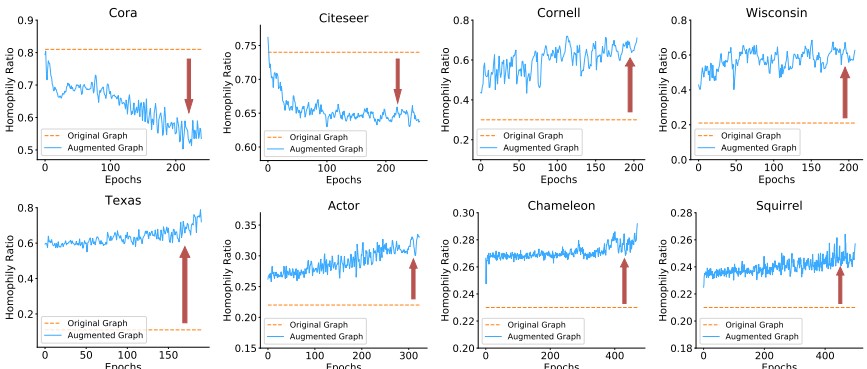

Figure 5: Training curves (w/o GKD Loss) of homophily ratios in the original and augmented graphs.

## 5.3 Ablation Study and Parameter Sensitivity

**Ablation on Student Model Designs.** The parameter-shared GNN shown in Fig. 3(c) is adopted as the student model by default in this paper for a fair comparison. In this subsection, we delve into the applicability of the proposed KDGA framework to different student model designs. Specifically, with the vanilla GCN as the base architecture and GraphAug as the graph augmentation method, we compare the performance of the

Table 2: Ablation study on student model designs.

| Method | Cora | Citeseer | Chameleon | Squirrel | Actor |
|---|---|---|---|---|---|
| Vanilla GCN | $81.5_{\pm0.8}$ | $71.6_{\pm0.3}$ | $59.8_{\pm2.6}$ | $36.9_{\pm1.3}$ | $30.3_{\pm0.8}$ |
| GraphAug | $83.2_{\pm0.9}$ | $73.2_{\pm0.8}$ | $58.8_{\pm1.8}$ | $37.2_{\pm1.2}$ | $28.8_{\pm0.9}$ |
| KDGA w/ Param-S | $85.2_{\pm0.7}$ | $73.9_{\pm0.7}$ | $62.7_{\pm1.5}$ | $46.9_{\pm0.6}$ | $32.5_{\pm0.6}$ |
| $\Delta_{Acc}$ | 2.0 | 0.7 | 3.9 | 9.7 | 3.7 |
| KDGA w/ Param-I | $84.0_{\pm0.6}$ | $72.7_{\pm0.5}$ | $60.6_{\pm1.7}$ | $40.5_{\pm1.0}$ | $30.7_{\pm0.9}$ |
| $\Delta_{Acc}$ | 0.8 | -0.5 | 1.8 | 3.3 | 1.9 |
| Vanilla MLP | $55.2_{\pm0.5}$ | $46.5_{\pm0.5}$ | $46.4_{\pm2.5}$ | $29.7_{\pm1.8}$ | $35.8_{\pm1.0}$ |
| KDGA w/ MLP | $83.2_{\pm1.1}$ | $73.5_{\pm0.7}$ | $58.1_{\pm1.0}$ | $38.8_{\pm0.7}$ | $38.1_{\pm0.8}$ |
| $\Delta_{Acc}$ | 28.0 | 27.0 | 11.7 | 9.1 | 2.3 |

parameter-shared model (w/ Param-S) in Fig. 3(c) and the parameter-independent model (w/ Param-I) in Fig. 3(b) on five datasets. It can be seen from Table. 2 that although the Param-I model can also improve the performance of GraphAug overall, it may fail on a few datasets, such as a 0.5% accuracy drop on Citeseer; more importantly, its performance gain falls far behind the Param-S model on all five datasets. The reason behind this may be that a parameter-independent model may be quickly fitted with the original graph structure while failing to take full advantage of the rich contextual information embedded in the augmented graphs. Moreover, we also consider a variant of the Param-I model by directly taking a parameter-independent MLP (w/ MLP) as the student mode. We find from Table. 2 that even with a simple MLP, it can still benefit from the augmented graphs and achieves performance beyond that of its vanilla implementations.

**Sensitivity Analysis on Hyperparameters.** We have evaluated the parameter sensitivity w.r.t two key hyperparameters: fusion factor $\alpha$ and loss weight $\kappa$. However, due to space limitations, we have placed the corresponding results of sensitivity analysis in **Appendix C**.

## 6 Conclusion

In this paper, we identified a potential negative augmentation problem for graph augmentation, which is caused by overly severe distribution shifts between the original and augmented graphs. To address this problem, we propose a novel _Knowledge Distillation for Graph Augmentation_ (KDGA) framework by directly distilling contextual information from a teacher model trained on the augmented graphs into a partially parameter-shared student model. Extensive experiments show that KDGA outperforms the vanilla implementations of existing augmentation methods and GNN architectures. Limitations still exist, such as KDGA requires an initial raw graph structure for augmentation and cannot be applied to those structure-unknown scenarios, which will be left for future work.

## 7 Acknowledgement

This work is supported in part by Ministry of Science and Technology of the People's Republic of China (No. 2021YFA1301603) and National Natural Science Foundation of China (No. U21A20427).

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
