# OpenReview forum: "Knowledge Distillation Improves Graph Structure Augmentation for Graph Neural Networks"
_NeurIPS.cc/2022/Conference — NeurIPS 2022 Accept_

### Official Review · Reviewer_LrXs · 2022-06-23

**Rating:** 7
**Confidence:** 4
**Soundness:** 3 good
**Presentation:** 3 good
**Contribution:** 2 fair

**Summary:**

The authors investigate on an interesting problem for graph augmentation: distribution shift between the original and the augmented graphs. The authors leveraged knowledge distillation to narrow the distribution gaps between the original and the augmented graphs. Experiment results show that the proposed framework (KDGA) based on the existing graph augmentation algorithms achieve superior results.

**Questions:**

See above

**Limitations:**

See above

**Strengths And Weaknesses:**

Strengths:
1. The authors pointed out the observation that there is a distribution shift between the original graph and the augmented graph. They also proposed to use the graph homophily ratio as the metric to quantify the distribution shift.
2. The authors illustrated the concepts and methods clearly with well presented figures. The overall presentation is good and easy to follow

Weakness:
The proposed the KDGA framework is based on the existing graph augmentation algorithms, and the student-teacher distillation framework. Would like to see more novelty in this work.

---

> ### Author Response · Authors · 2022-08-01
> **Response to Reviewer LrXs - Part 1**
>
> Thanks for your insightful reviews, and we appreciate your valuable suggestions! We address your concerns and questions as follows:
>
> Q1:  We would like to see more novelty in this work.
>
> A1: We would like to emphasize that KDGA is not a simple stacking of existing knowledge distillation and graph augmentation methods, which completely overlooks our contributions. To highlight our innovation, we re-summarize our contribution as follows:
>
> -  **Problem Identification.** Up to the submission of this manuscript to NIPS2022, we are the first to identify the potential negative augmentation problem. Moreover, we have described in detail what it represents, how it arises, what impact it has, how it behaves on real-world data, and how it can be solved. The negative augmentation is a common problem among existing graph augmentation methods, and we believe that **the identification and in-depth exploration of this problem are far more valuable than developing a new graph augmentation method**.
> -  **A knowledge distillation-based solution.** We provide a solution to the identified problem, where we do not directly prevent the occurrence of distribution shifts, but try to reduce their negative effects (i.e., potential negative augmentation) through knowledge distillation.
> -  Please note that **KDGA is a general framework**, rather than a specific method. In practice, any existing graph augmentation method can be combined with the KDGA framework and achieve consistent improvements over the vanilla baselines, as shown in Table 1. As a small bonus, we also propose a novel probabilistic generative-based graph augmentation method, called GraphAug, and use it to instantiate KDGA in Section 4.4.
>
> ---
>
> In light of these responses, we hope we have addressed your concerns. If we have left any notable points of concern unaddressed, please do share and we will attend to these points. We sincerely hope that you can appreciate our efforts on responses and revisions and thus raise your score.

---

### Official Review · Reviewer_SAR2 · 2022-07-11

**Rating:** 7
**Confidence:** 4
**Soundness:** 3 good
**Presentation:** 3 good
**Contribution:** 3 good

**Summary:**

This paper claims that they are the first in considering the distributional shift between an original and augmented graph. They propose Knowledge Distillation for Graph Augmentation (KDGA) to address the potentially negative impact of this distributional shift. The model consists of a teacher and a student model. The teacher is a GNN model that extracts information from the augmented graph and injects the knowledge into the student model which has only seen the original graph by minimizing the KL divergence of the output logits of these models.

**Questions:**

-- Have the authors studied the sensitivity of the final performance on the GKD loss weight in equation 9?

-- Figure 5 is showing homophily ratios in the original and augmented graphs during training w/o the GKD loss, right? If that's true, what would the trajectory look like with the GKD loss?

**Limitations:**

there is not social impact section in the manuscript.

**Strengths And Weaknesses:**

**Strength**

-- The paper is well-written and the authors are trying to tackle an important aspect of graph contrastive learning.

-- The approach is simple and can be easily implemented in an efficient way.

**Weaknesses**

-- The authors should make it clear from the begining of the manuscript that by 'distribution' they mean the homophily ratios of the graph structures.

-- The claim in line 63 that little work studies structure augmentation is not accurate just by considering the already cited literature.

-- The claim in lines 130 and 131 that graph structures for training and testing in graph augmentations are completely different isn't necessarily true. This is a property you can control in a few graph contrastive methods and automated GCL methods might limit this issue automatically.

-- typo: Figure 5 caption, "homopgily"

---

> ### Author Response · Authors · 2022-08-01
> **Response to Reviewer SAR2 - Part 1**
>
> Thanks for your insightful reviews, and we appreciate your valuable suggestions! We address your concerns and questions as follows:
>
> Q1: The authors should make it clear from the beginning of the manuscript that by 'distribution' they mean the homophily ratios of the graph structures.
>
> A1: We have made it clear in the abstract in Lines 7-9 that we take the homophily ratio of the graph to analyze (or measure) distribution shifts. Moreover, we have discussed in detail in Lines 100-107 what the graph homophily represents and why we use it to measure distributional shifts.
>
> ---
>
> Q2: The claim in line 63 that little work studies structure augmentation is not accurate just by considering the already cited literature.
>
> A2: We are quite confused by your comment, and could you please make it more clear? We respond to this comment with the following two points:
>
> -  **Presentation.** The statement " comparatively little work study graph (structure) augmentation" is made relative to node feature augmentation, and we have never claimed that graph structure augmentation is a completely blank research field.
> -  **Related Work.** We guess what you want to argue is that we did not discuss and cite those augmentation methods used for graph contrastive learning. However, we have specifically distinguished graph structure augmentation from Graph Contrast Learning (GCL) in Lines 74-77 and Lines 83-89 in the original manuscript, as their learning objectives and evaluation protocols are completely different. For the task of graph (structure) augmentation, we are convinced that our cited literature is reasonable. If you still assert that our cited literature is inadequate, please list out the missing work to help improve the quality of our manuscript, but please don't refer to those augmentation methods tailored for GCL.
>
> ---
>
> Q3: The claim in lines 130 and 131 that graph structures for training and testing in graph augmentations are completely different isn't necessarily true. This is a property you can control in a few graph contrastive methods and automated GCL methods might limit this issue automatically.
>
> A3: The answers to Q3 include the following three aspects:
>
> - We are quite familiar with graph contrastive learning, including several adaptive augmentation methods [1,2,3] and automated GCL [3,4,5], and have published several works on GCL. However, we would like to emphasize again that graph structure augmentation and GCL have completely different learning objectives and evaluation protocols, as discussed in Lines 83-89, so please do not simply consider our approach as just one aspect of GCL.
>
> - The role of graph structure perturbation in GCL is to generate **different views as positive/negative samples** and then perform contrasting in an **unsupervised** manner to learn transferable knowledge. In contrast, the purpose of graph structure augmentation in this paper is to reasonably perturb the graph structure to enable nodes to **receive richer contextual information**, usually working in a **semi-supervised** setting. To avoid any possible misunderstanding, we have specifically discussed how graph structure augmentation differs from GCL in Lines 74-77 and Lines 83-89.
>
> - We strongly encourage you to re-read Section 4.1 and refer to Figure 1 to understand exactly what graph structure augmentation stands for, and why we say that "the graph structures for training and testing are completely different." Besides, we also disagree with your claim that our work is the same thing as the "property" you defined for graph contrastive learning (BTW, "property" is a rather vague term, what exactly does it refer to?).
>
> ---
>
> Q4:  typo: Figure 5 caption, "homopgily"
>
> A4: All these typos, including but not limited to grammatical errors and misspellings of words, have been corrected in the revised manuscript.
>
> ---
>
> Q5:  Have the authors studied the sensitivity of the final performance on the GKD loss weight in equation 9?
>
> A5: The parameter sensitivity w.r.t the GKD loss weight $\kappa$ has been added in Appendix D (Figure A2(b) and Lines 568-576). In practice, we find that setting $\kappa$ to a non-zero value, i.e., training with GKD loss, always achieves better performance than that of training without GKD loss (i.e., setting $\kappa=0$), which demonstrates the effectiveness of the GKD loss. Moreover, we find that the model performance can be further improved with increasing loss weights $\kappa$, but the performance gains reduce when $\kappa$ becomes too large, possibly because a too large GKD loss weight $\kappa$ tends to weaken the contribution of label information (i.e., loss of $\mathcal{L}_{cla}$) in the semi-supervised setting. More sensitivity analysis on GKD loss weight $\kappa$  is available in Lines 568-576.

---

> > ### Author Response · Authors · 2022-08-01
> > **Response to Reviewer SAR2 - Part 2**
> >
> > Q6:  Figure 5 shows homophily ratios during training w/o the GKD loss; what would the trajectory look like with the GKD loss?
> >
> > A6: We have provided the trajectory (with the GKD loss) in  Figure A1, discussed the role of GKD loss in Lines 546-553, and made corresponding changes in Lines 140-143 and Lines 159-165, to help readers grasp the main points. The answers to Q6 include the following three aspects:
> >
> > - **Additional results.** The training curves (trained w/ and w/o GKD Loss) of the homophily ratios have been provided in Figure 5 and Figure A1.
> > - **Discussion on the role of GKD loss.** The GKD loss serves as a "bridge" between the teacher and student models. It regularizes the student model by gradually distilling knowledge from a teacher model pre-trained on augmented graphs to a student model, but does not directly affect the learning of the teacher model and graph augmentation. As a result, the trajectories trained with and without GKD loss will not be substantially different, i.e., they both still suffer from distribution shifts, as shown in Figure 5 and Figure A1. Essentially, the role of GKD loss is not to directly prevent the occurrence of distribution shifts, but to reduce their negative effects (i.e., potential negative augmentation) as much as possible, as shown in Table 1.
> > - **Major changes in the revised manuscript.** To help readers grasp the main points, we have specifically added two paragraphs in Lines 140-143 and Lines 159-165 to discuss the positive/negative effects of distribution shifts and the main ideas of this paper. In essence, the distribution shift itself is not necessarily harmful; it is essentially a trade-off between better generalizability and higher risks of negative augmentation.
> >
> > ---
> >
> > In light of these responses, we hope we have addressed your concerns. If we have left any notable points of concern unaddressed, please do share and we will attend to these points. We sincerely hope that you can appreciate our efforts on responses and revisions and thus raise your score.
> >
> > ---
> >
> > [1] Zhu Y, Xu Y, Yu F, et al. Graph contrastive learning with adaptive augmentation[C]//Proceedings of the Web Conference 2021. 2021: 2069-2080.
> >
> > [2] Suresh S, Li P, Hao C, et al. Adversarial graph augmentation to improve graph contrastive learning[J]. Advances in Neural Information Processing Systems, 2021, 34: 15920-15933.
> >
> > [3] Liu S, Ying R, Dong H, et al. Local augmentation for graph neural networks[C]//International Conference on Machine Learning. PMLR, 2022: 14054-14072.
> >
> > [4] You Y, Chen T, Shen Y, et al. Graph contrastive learning automated[C]//International Conference on Machine Learning. PMLR, 2021: 12121-12132.
> >
> > [5] Yin Y, Wang Q, Huang S, et al. Autogcl: Automated graph contrastive learning via learnable view generators[C]//Proceedings of the AAAI Conference on Artificial Intelligence. 2022, 36(8): 8892-8900.
> >
> > [6] Jin W, Liu X, Zhao X, et al. Automated self-supervised learning for graphs[J]. arXiv preprint arXiv:2106.05470, 2021.

---

> ### Author Response · Authors · 2022-08-08
> **Rebuttal has been submitted and we are sincerely looking forward to your reply**
>
> Dear Reviewer,
>
> We appreciate your time and effort in reviewing this paper. We tried our best to address all of the aforementioned concerns/issues. If you still have any questions, please feel free to contact us.
>
> Best, Authors

---

> > ### Comment · Reviewer_SAR2 · 2022-08-08
> > **Thanks for the response**
> >
> > I read the authors' responses and quickly reviewed others' comments.
> >
> > Although there are still some minor misunderstandings in our discussion (especially regarding GCL) I think the authors addressed most of the concerns.
> >
> > I'm increasing my score to 7.

---

> > > ### Author Response · Authors · 2022-08-09
> > > **Thanks for increasing your score**
> > >
> > > We' re glad to hear that we have addressed most of your concerns! Thanks for spending a large amount of time on our submission, which makes our paper even stronger.

---

### Official Review · Reviewer_o5A2 · 2022-07-13

**Rating:** 5
**Confidence:** 4
**Soundness:** 2 fair
**Presentation:** 3 good
**Contribution:** 2 fair

**Summary:**

This paper claims that the existing graph structure augmentation methods may cause the distribution shift issue. In this paper, the authors design a knowledge distillation based method to improve the generalization capability for graph structure augmentation.

**Questions:**

See the weaknesses.

**Ethics Review Area:**

["I don’t know"]

**Limitations:**

N.A.

**Strengths And Weaknesses:**

Strengths:
1. Authors find that the existing graph augmentation method will lead to the distribution shift problem.
2. This paper is well-written and easy to follow
3. The idea behind is straightforward.

Weaknesses:
1. The authors claim that Negative Augmentation will lead to poor generalization. While It does not make sense based on your experiments.  The experiments show that the existing methods do perform well than baselines. This means that if those baselines are negative augmentation, they should perform worse. Or authors may need to conduct the experiment specifically to test the generalization.
2. While the existing methods have a big gap in homopgily ratio from the augmented to original data, but no experiments show that the proposed method does not have it. Moreover, why big gap in homopgily atio represents worse generalization capability? intuitively, a big gap in homopogily ratio can help model to ''see'' more different graphs and increase the generalization.
3 . Typos in Table 2: GKDA -> KDGA?
4. The novelty is limited. As the proposed KDGA is just a simple knowledge Distillation method. The authors just apply it to existing augmentation methods to achieve better performance.

---

> ### Author Response · Authors · 2022-08-01
> **Response to Reviewer o5A2 - Part 1**
>
> Thanks for your insightful reviews, and we appreciate your valuable suggestions! We address your concerns and questions as follows:
>
> Q1: The authors claim that Negative Augmentation will lead to poor generalization. While it does not make sense based on your experiments. The experiments show that the existing methods do perform well than baselines. If those baselines are negative augmentation, they should perform worse.
>
> A1: We first clarify our opinions on the roles of negative augmentation, then present the changes we have made, and finally analyze it with corresponding  experimental results. The answers to Q1 include the following four aspects:
>
> - We never claimed that existing graph augmentation methods necessarily perform worse than vanilla baselines due to suffering from negative augmentation. What we really mean is that while existing graph augmentation methods have demonstrated their effectiveness on a few datasets, the potential negative augmentation problem (as analyzed in Section 4.2) may be ***an obstacle to hinder further improvement*** in their performance.
> - To avoid possible misunderstanding, we have rewritten the expression from "negative augmentation will lead to poor generalization" to "negative augmentation will lead to **suboptimal** generalization" in Lines 47-48 in the revised manuscript.
> - Our experimental results, as shown in Table 1, also do not violate our statements at all. For example, although all three augmentation methods, GAUG, MH-Aug, and GraphAug, help to improve performance on homophily datasets, their performance can be further improved by combining them with our KDGA framework, as shown in Table 1. Moreover, on six heterophily datasets, these methods lag behind those heuristic methods and are even worse than the vanilla baselines; however, by combining KDGA, their performance can be greatly improved by up to 9.9%. Overall, KDGA improves the vanilla GNN baselines by 4.6% (GAUG), 4.2% (MH-Aug), and 4.6% (GraphAug) on eight public datasets, respectively. The above analysis and discussion on the experimental results of Table 1 have been provided in Lines 249-256 of the revised manuscript.
>
> ---
>
> Q2: While the existing methods have a big gap in homophily ratio from the augmented to original data, but no experiments show that the proposed method does not have it. Moreover, why a big gap in the homophily ratio represents worse generalization capability? Intuitively, a big gap in the homophily ratio can help the model to ''see'' more different graphs and increase the generalization.
>
> A2: We first discuss the roles played by distribution shifts in graph augmentation, then clarify the main idea of this paper, i.e., how we deal with distribution shifts, and finally present the changes we have made in the revised manuscript. The answers to Q2 include the following four aspects:
>
> - **Roles of distribution shifts.** In essence, the distribution shift itself is not necessarily harmful; it is actually a neutral phenomenon. A proper distribution shift helps the model to "see" more different graphs, enabling the nodes to receive more contextual information and thus improve generalization. However, this does not suggest that a larger homology ratio gap is necessarily better, as it entails an overly severe distribution shift, which can significantly increase the risk of the negative augmentation problem, as analyzed in Section 4.2.
> - **Our Solution.** The distribution shift is essentially a trade-off between better generalizability and higher risks of negative augmentation. However, the optimal distribution shift may vary from dataset to dataset, or even from node to node, making it challenging to directly control the levels of distribution shifts. In this paper, we have not attempted to control or prevent distribution shifts. Instead, we allow for the existence of any level of distribution shifts, but we reduce their negative impact, i.e., the potential negative augmentation problem, by the proposed KDGA framework.
> - **Experimental results.** The training curves (trained w/ and w/o GKD Loss) of the homophily ratios have been provided in Figure 5 and Figure A1, respectively, from which we find that they both suffer from distribution shifts. However, while the role of GKD loss is not to directly prevent the occurrence of distribution shifts, they help to reduce their negative effects (potential negative augmentation)  and consistently improve the performance of existing graph augmentation methods, as shown in Table 1.  For example, KDGA improves the vanilla GNN baselines by an average accuracy of 4.6% (GAUG), 4.2% (MH-Aug), and 4.6% (GraphAug) on eight public datasets, respectively.
> - **Changes in the manuscript.**  To help readers grasp the main points, we have added two paragraphs to discuss the positive and negative effects of distribution shifts in Lines 140-143 and how we deal with their negative effects (potential negative augmentation) in Lines 159-165.

---

> > ### Author Response · Authors · 2022-08-01
> > **Response to Reviewer o5A2 - Part 2**
> >
> > Q3: 3 . Typos in Table 2: GKDA -> KDGA?
> >
> > A3: All these typos, including but not limited to grammatical errors and misspellings of words, have been corrected in the revised manuscript.
> >
> > ---
> >
> > Q4:  The novelty is limited.
> >
> > A4: We would like to emphasize that KDGA is not a simple stacking of existing knowledge distillation and graph augmentation methods, which completely overlooks our contributions. To highlight our innovation, we re-summarize our contribution as follows:
> >
> > -  **Problem Identification.** Up to the submission of this manuscript to NIPS2022, we are the first to identify the potential negative augmentation problem. Moreover, we have described in detail what it represents, how it arises, what impact it has, how it behaves on real-world data, and how it can be solved. The negative augmentation is a common problem among existing graph augmentation methods, and we believe that **the identification and in-depth exploration of this problem are far more valuable than developing a new graph augmentation method**.
> > -  **A knowledge distillation-based solution.** We provide a solution to the identified problem, where we do not directly prevent the occurrence of distribution shifts, but try to reduce their negative effects (i.e., potential negative augmentation) through knowledge distillation.
> > -  Please note that **KDGA is a general framework**, rather than a specific method. In practice, any existing graph augmentation method can be combined with the KDGA framework and achieve consistent improvements over the vanilla baselines, as shown in Table 1. As a small bonus, we also propose a novel probabilistic generative-based graph augmentation method, called GraphAug, and use it to instantiate KDGA in Section 4.4.
> >
> > ---
> >
> > In light of these responses, we hope we have addressed your concerns. If we have left any notable points of concern unaddressed, please do share and we will attend to these points. We sincerely hope that you can appreciate our efforts on responses and revisions and thus raise your score.

---

> > > ### Comment · Reviewer_o5A2 · 2022-08-09
> > > **Most of my concerns are addressed. Thanks for the efforts.**
> > >
> > > Most of my concerns are addressed. Thanks for the efforts. I raised my score to borderline accept.

---

> > > > ### Author Response · Authors · 2022-08-09
> > > > **Thanks for increasing your score**
> > > >
> > > > We' re glad to hear that we have addressed most of your concerns! Thanks for spending a large amount of time on our submission, which makes our paper even stronger.

---

> ### Author Response · Authors · 2022-08-08
> **Rebuttal has been submitted and we are sincerely looking forward to your reply**
>
> Dear Reviewer,
>
> We appreciate your time and effort in reviewing this paper. We tried our best to address all of the aforementioned concerns/issues. If you still have any questions, please feel free to contact us.
>
> Best, Authors

---

### Official Review · Reviewer_J4nz · 2022-07-15

**Rating:** 5
**Confidence:** 4
**Soundness:** 3 good
**Presentation:** 4 excellent
**Contribution:** 3 good

**Summary:**

The authors of the paper identify and demonstrate a negative augmentation problem in graph augmentation methods. They propose to use a knowledge distillation method to solve this problem by training a teacher model on the augmented graphs and a student model on the original graphs. Experiments are conducted on two homogeneous graph datasets and six heterogeneous graph datasets. Improved performance are observed when the knowledge distillation method is applied to three learning-based graph augmentation methods and three different GNNs.

**Questions:**

1. Can other techniques achieve the same effect, such as increasing regularization, increasing the diversity of the sampled augmented graphs? Could you provide a comparison with these methods?
2. Has anyone proposed using knowledge distillation (in any way) to solve the negative augmentation problem?
3. Can KGDA be easily applied to graph contrastive learning or any other graph application?

**Limitations:**

1. Evaluations are only conducted on the node classification task. Would like to see experiments on more tasks, such as edge classification, graph contrastive learning.
2. The proposed method does not seem to address the problem that graph augmentation methods tend to increase the homophily ratio of the graphs. It is more of a regularization that guides the student model on downstream tasks.

**Strengths And Weaknesses:**

Strength
1. The authors demonstrated sound statistics and analysis of the negative augmentation problem.
2. Extensive experimental results are presented on the node classification task, using three representative GNNs and three learning-based graph augmentation methods.
3. The paper is well written.

Weakness
1. In Figure 3, are the inputs of the teacher model and the student model swapped?
2. Evaluations are only conducted on the node classification task. Would like to see experiments on more tasks, such as edge classification, graph contrastive learning.

---

> ### Author Response · Authors · 2022-08-01
> **Response to Reviewer J4nz - Part 1**
>
> Thanks for your insightful reviews, and we appreciate your valuable suggestions! We address your concerns and questions as follows:
>
> Q1: In Figure 3, are the inputs of the teacher model and the student model swapped?
>
> A1: We have swapped the inputs of the student and teacher models in Figure 3, and it now presents the right picture.
>
> ---
>
> Q2: Evaluations are only conducted on the node classification task. Could you provide experiments for more tasks, such as edge classification and graph contrastive learning?
>
> A2: We first explain why Graph Contrastive Learning (GCL) and edge classification are not reasonable tasks for evaluating graph augmentation, and then present our reasons for focusing on semi-supervised node classification, from the following four aspects:
>
> - **Graph contrastive learning (GCL).** Please note that Graph Contrastive Learning (GCL) is a learning framework (method), rather than a special type of graph-related task. It is a factual error to consider GCL as a task. This paper has discussed in detail in Lines 83-89 what are the essential differences between graph (structure) augmentation and GCL, from the perspectives of both learning objectives and evaluation protocols.
> - **Edge classification.** We have described in detail the statistical characteristics of the eight datasets used, and in particular how they are constructed, in Lines 512-528 in Appendix A. Apart from edge connectivity, we know very little about any ground-truth edge labels or edge attributes, which prevents us from conducting experiments on the task of edge classification.
> - **Node classification.** There is no *optimal* augmentation method that is universal for all data and task types. In recent years, the emphasis has been increasingly on task-specific augmentation; for example, there are [1] for temporal graph augmentation, [2,3] for the task of graph classification, and [4,5] for the task of knowledge graph completion. The graph (structure) augmentation is also tailored for node-level classification tasks, because **nodes (rather than edges or graphs) are more likely to benefit from structure perturbations** by capturing richer  contextual information.
> - KDGA is not a specific method, and it is a general framework, which in practice can be combined with any existing graph augmentation method. Therefore, **whether KDGA is applicable to other tasks depends entirely on whether these augmentation methods can do it.** Unfortunately, however, almost all existing graph augmentation methods, including GAUG [6], MH-Aug [7], and DropEdge [8], all focus on node-level classification tasks and cannot directly generalize to other tasks. To make a fair comparison, this paper only provides comparisons for the node classification task.
>
> ---
>
> Q3: Can other techniques achieve the same effect, such as increasing regularization or increasing the diversity of the sampled augmented graphs? Could you provide a comparison with these methods?
>
> A3: We first discuss what constitutes a fair comparison for our work, and then add comparisons with new baselines, from the following two aspects:
>
> - **Fair comparison.** KDGA is not a specific method, and it is a general framework, which in practice can be combined with any existing graph augmentation method. Therefore, in a fair comparison, the effectiveness of KDGA should be evaluated by **how much it improves existing graph augmentation**, including GAUG, MH-Aug, and GraphAug, rather than directly comparing it to alternatives (or competitors) of graph augmentation.
> - **Results and Analysis.** There are also some semi-supervised methods aiming to improve the generalization of graph training, such as SSL [9] and GraphMix [10]. We have added a discussion about them in Lines 236-239, and an experimental comparison with them in Table 1 and Line 259-262.
>
> ---
>
> Q4:  Has anyone proposed using knowledge distillation (in any way) to solve the negative augmentation problem?
>
> A4: Up to the submission of this manuscript to NIPS2022, we are the first to identify the negative augmentation problem; we describe in detail what it represents, how it arises, what impact it has, and how it can be solved by knowledge distillation. To the best of our knowledge, there is no readily available method for this negative augmentation problem. More solutions to the negative augmentation problem (including but not limited to knowledge distillation-style solutions) will be left to later researchers as future work.

---

> > ### Author Response · Authors · 2022-08-01
> > **Response to Reviewer J4nz - Part 2**
> >
> > Q5:  The proposed method does not seem to address the problem that graph augmentation methods tend to increase the homophily ratio of the graphs.
> >
> > A5: We first discuss the roles played by distribution shifts in graph augmentation, then clarify the main idea of this paper, i.e., how we deal with distribution shifts, and finally present the changes we have made in the revised manuscript. The answers to Q5 include the following four aspects:
> >
> > - **Roles of distribution shifts.** In essence, the distribution shift itself is not necessarily harmful; it is actually a neutral phenomenon. A proper distribution shift helps the model to "see" more different graphs, enabling the nodes to receive more contextual information and thus improve generalization. However, this does not suggest that a larger homology ratio gap is necessarily better, as it entails an overly severe distribution shift, which can significantly increase the risk of the negative augmentation problem, as analyzed in Section 4.2.
> > - **Our Solution.** The distribution shift is essentially a trade-off between better generalizability and higher risks of negative augmentation. However, the optimal distribution shift may vary from dataset to dataset, or even from node to node, making it challenging to directly control the levels of distribution shifts. In this paper, we have not attempted to control or prevent distribution shifts. Instead, we allow for the existence of any level of distribution shifts, but we reduce their negative impact, i.e., the potential negative augmentation problem, by the proposed KDGA framework.
> > - **Experimental results.** The training curves (trained w/ and w/o GKD Loss) of the homophily ratios have been provided in Figure 5 and Figure A1, respectively, from which we find that they both suffer from distribution shifts. However, while the role of GKD loss is not to directly prevent the occurrence of distribution shifts, they help to reduce their negative effects (potential negative augmentation)  and consistently improve the performance of existing graph augmentation methods, as shown in Table 1.  For example, KDGA improves the vanilla GNN baselines by an average accuracy of 4.6% (GAUG), 4.2% (MH-Aug), and 4.6% (GraphAug) on eight public datasets, respectively.
> > - **Changes in the manuscript.**  To help readers grasp the main points, we have added two paragraphs to discuss the positive and negative effects of distribution shifts in Lines 140-143 and how we deal with their negative effects (potential negative augmentation) in Lines 159-165.
> >
> > ---
> >
> > In light of these responses, we hope we have addressed your concerns. If we have left any notable points of concern unaddressed, please do share and we will attend to these points. We sincerely hope that you can appreciate our efforts on responses and revisions and thus raise your score.
> >
> > ---
> >
> > [1] Wang Y, Cai Y, Liang Y, et al. Adaptive data augmentation on temporal graphs[J]. Advances in Neural Information Processing Systems, 2021, 34: 1440-1452.
> >
> > [2] Zhou J, Shen J, Yu S, et al. M-evolve: structural-mapping-based data augmentation for graph classification[J]. IEEE Transactions on Network Science and Engineering, 2020, 8(1): 190-200.
> >
> > [3] Zhou J, Shen J, Xuan Q. Data augmentation for graph classification[C]//Proceedings of the 29th ACM International Conference on Information & Knowledge Management. 2020: 2341-2344.
> >
> > [4] Tang Z, Pei S, Zhang Z, et al. Positive-Unlabeled Learning with Adversarial Data Augmentation for Knowledge Graph Completion[J]. arXiv preprint arXiv:2205.00904, 2022.
> >
> > [5] Chauhan J, Gupta P, Minervini P. A Probabilistic Framework for Knowledge Graph Data Augmentation[J]. arXiv preprint arXiv:2110.13205, 2021.
> >
> > [6] Zhao T, Liu Y, Neves L, et al. Data augmentation for graph neural networks[C]//Proceedings of the AAAI Conference on Artificial Intelligence. 2021, 35(12): 11015-11023.
> >
> > [7] Park H, Lee S, Kim S, et al. Metropolis-hastings data augmentation for graph neural networks[J]. Advances in Neural Information Processing Systems, 2021, 34: 19010-19020.
> >
> > [8] Rong Y, Huang W, Xu T, et al. Dropedge: Towards deep graph convolutional networks on node classification[J]. arXiv preprint arXiv:1907.10903, 2019.
> >
> > [9] Qikui Zhu, Bo Du, and Pingkun Yan. Self-supervised training of graph convolutional networks. arXiv preprint arXiv:2006.02380, 2020.
> >
> > [10] Vikas Verma, Meng Qu, Kenji Kawaguchi, Alex Lamb, Yoshua Bengio, Juho Kannala, and Jian Tang. Graphmix: Improved training of gnns for semi-supervised learning. In Proceedings of the AAAI Conference on Artificial Intelligence, volume 35, pages 10024–10032, 2021.

---

> > > ### Comment · Reviewer_J4nz · 2022-08-09
> > > **Thanks the authors for your detailed reply. Most of my concerns have been addressed and I maintain my original rating.**
> > >
> > > Thanks the authors for your detailed reply. Most of my concerns have been addressed and I maintain my original rating.

---

> > > > ### Author Response · Authors · 2022-08-09
> > > > **Glad to hear that most of your concerns have been addressed and you are willing to keep a positive rating score.**
> > > >
> > > > We' re glad to hear that we have addressed most of your concerns and that you are still willing to keep a positive rating score! Thanks for spending a large amount of time on our submission, which makes our paper even stronger.

---

> ### Author Response · Authors · 2022-08-08
> **Rebuttal has been submitted and we are sincerely looking forward to your reply**
>
> Dear Reviewer,
>
> We appreciate your time and effort in reviewing this paper. We tried our best to address all of the aforementioned concerns/issues. If you still have any questions, please feel free to contact us.
>
> Best, Authors

---

### Author Response · Authors · 2022-08-01
**Summary of the major revision**

We thank the reviewers for their thorough and detailed reviews on our submission. We have incorporated the necessary changes in the revised manuscript. All changes are marked in red in the updated submission. The major changes are:

- We have revised and polished the paper and specifically discussed some issues that reviewers are concerned about, including:
  - **Comparison with Graph Contrastive Learning (GCL).** The essential differences between graph (structure) augmentation and GCL have been discussed in Lines 83-89 to avoid reviewers from simply treating graph (structure) augmentation as one aspect of GCL.
  - **Comparison with graph distillation.** We have discussed in Figure 3 and Lines 189-203 how GKDA differs from existing graph knowledge distillation, and highlighted again that KDGA is not a simple stacking of existing knowledge distillation and graph augmentation methods.
  - **Discussion on distribution shifts.** We have summarized our motivations on distribution shifts in Lines 140-143 and Lines 159-165; we have not attempted to directly prevent or completely eliminate distribution shifts. Instead, we allow the existence of any level of distribution shifts, but we reduce their negative impact, i.e., the potential negative augmentation problem, through the proposed KDGA framework.
- We have added some new experiments and conducted more detailed and in-depth analysis on some existing experimental results, including:
  - **Parameter sensitivity.** The parameter sensitivity w.r.t the GKD loss weight $\kappa$  has been provided in Figure A2(b) and Lines 568-576.
  - **Comparisons with new baselines.** We have added comparisons with two new semi-supervised baselines, SSL [1] and GraphMix [2], in Table 1, Lines 236-239, and Lines 259-262 to demonstrate the effectiveness of the proposed KDGA framework.
  - **Additional training curves.** The training curves (trained w/ and w/o GKD Loss) of the homophily ratios have been provided in Figure 5 and Appendix C (Figure A1 and Lines 546-553) in the revised manuscript to clarify that the role of GKD Loss is to serve as a "bridge" between the teacher and student models, rather than directly preventing the occurrence of distribution shifts.
  - **In-depth performance analysis.** The experimental analysis has been provided in Lines 249-256 to support our claim that "potential negative augmentation may lead to suboptimal generalization"; by combining with the proposed KDGA framework, the performance of existing graph augmentation methods, including GAUG, MH-Aug, and GraphAug, can be greatly improved across various datasets and GNN architectures.
- We would like to emphasize that KDGA is not a simple stacking of existing knowledge distillation and graph augmentation methods, which completely overlooks our contributions. To highlight our innovation, we re-summarize our contribution as follows:
  -  **Problem Identification.** Up to our submission to NIPS2022, we are the first to identify the potential negative augmentation problem. Moreover, we have described in detail what it represents, how it arises, what impact it has, how it behaves on real-world data, and how it can be solved. The negative augmentation is a common problem among existing graph augmentation methods, and we believe that **the identification and in-depth exploration of this problem are far more valuable than developing a new graph augmentation method**.
  -  **A knowledge distillation-based solution.** We provide a solution to the identified problem, where we do not directly prevent the occurrence of distribution shifts, but try to reduce their negative effects (i.e., potential negative augmentation) through knowledge distillation.
  -  **KDGA is a general framework**, rather than a specific method. In practice, any existing graph augmentation method can be combined with the KDGA framework and achieve consistent improvements over the vanilla baselines, as shown in Table 1. As a small bonus, we also propose a novel probabilistic generative-based graph augmentation method, called GraphAug, and use it to instantiate KDGA in Section 4.4.

---

[1] Qikui Zhu, Bo Du, and Pingkun Yan. Self-supervised training of graph convolutional networks. arXiv preprint arXiv:2006.02380, 2020.

[2] Vikas Verma, Meng Qu, Kenji Kawaguchi, Alex Lamb, Yoshua Bengio, Juho Kannala, and Jian Tang. Graphmix: Improved training of gnns for semi-supervised learning. In Proceedings of the AAAI Conference on Artificial Intelligence, volume 35, pages 10024–10032, 2021.

---

### Author Response · Authors · 2022-08-07
**Rolling Discussion**

Dear Reviewers,

We appreciate your time and effort in reviewing this paper. We tried our best to address all of the aforementioned concerns/issues. If there is still anything that is not explained clearly in our response, we could elucidate them more.

Best, Authors

---

### Meta-Review · Area_Chair_wLda · 2022-08-28

**Recommendation:** Accept
**Confidence:** Certain

**Metareview:**

The paper identifies the problem of negative augmentation in graph augmentation methods as it may cause the distribution shift issue. The paper thus proposes a knowledge distillation method that trains a teacher model on the augmented graphs and a student model on the original one. Reviewers had concerns on the novelty of the approach and experiments. The discussion between the reviewers and the authors were effective, and two of the reviewers have raised the scores to accept/borderline accept. I'd recommend acceptance.

**Award:**

No

---

### Decision · Program_Chairs · 2022-09-14

Accept